# Investigating the Migraine Cycle over 21 Consecutive Days Using Proton Magnetic Resonance Spectroscopy and Resting-State fMRI: A Pilot Study

**DOI:** 10.3390/brainsci12050646

**Published:** 2022-05-14

**Authors:** Vera Filippi, Ruth Steiger, Vincent Beliveau, Florian Frank, Katharina Kaltseis, Elke R. Gizewski, Gregor Broessner

**Affiliations:** 1Department of Neurology, Innsbruck Medical University, 6020 Innsbruck, Austria; vera.filippi@i-med.ac.at (V.F.); vincent.beliveau@i-med.ac.at (V.B.); f.frank@i-med.ac.at (F.F.); katharina.kaltseis@i-med.ac.at (K.K.); 2Neuroimaging Research Core Facility, Innsbruck Medical University, 6020 Innsbruck, Austria; ruth.steiger@i-med.ac.at; 3Department of Neuroradiology, Innsbruck Medical University, 6020 Innsbruck, Austria; elke.gizewski@i-med.ac.at

**Keywords:** 1H-MRS, resting-state fMRI, migraine cycle, attacks

## Abstract

Recent neuroimaging studies have revealed important aspects of the underlying pathophysiological mechanisms of migraine suggesting abnormal brain energy metabolism and altered functional connectivity. Proton magnetic resonance spectroscopy (1H-MRS) studies investigated migraine patients in the interictal or ictal state. This first-of-its-kind study aimed to investigate the whole migraine cycle using 1H-MRS and resting-state functional magnetic resonance imaging (fMRI). A migraine patient underwent 1H-MRS and resting-state fMRI for 21 consecutive days, regardless of whether he was in an interictal or ictal state. Metabolite ratios were assessed and compared to the intrinsic connectivity of subcortical brain areas. Probable migraine phase-dependent changes in N-acetyl aspartate (NAA)/total creatine (tCr) and choline (Cho)/tCr levels are found in the left occipital lobe and left basal ganglia. NAA reflects neuronal integrity and Cho cellular membrane turnover. Such abnormalities may increase the susceptibility to excitatory migraine triggers. Functional connectivity between the right hippocampus and right or left pallidum was strongly correlated to the NAA/Cho ratio in the right thalamus, suggesting neurochemical modulation of these brain areas through thalamic connections. To draw statistically significant conclusions a larger cohort is needed.

## 1. Introduction

The diagnosis of migraine, despite the high global prevalence of 11.6% [1], chronicity, and consequent disability, is based on a set of signs and symptoms defined in the International Committee of Headache Disorders, 3rd edition (ICHD-3) [2]. No robust diagnostic biomarkers have been established yet. The identification of such biomarkers may improve our understanding of the neurobiological mechanisms of migraine and open new avenues to provide adequate therapy for patients. Imaging techniques have revealed important aspects of the underlying pathophysiological mechanisms of migraine [3]. Resting-state functional magnetic resonance imaging (fMRI) studies have shown altered connectivity in the migraine brain, both in the interictal and the ictal states. Previous evidence suggests that changes in hypothalamic activity precede the onset of pain by up to 24 h [4]. A recent resting-state fMRI study demonstrated increased functional connectivity between the right thalamus and contralateral cortical areas responsible for pain modulation (e.g., the insular and orbitofrontal cortex) and decreased functional connectivity to ipsilateral cortical areas responsible for pain processing (e.g., primary somatosensory cortex) [5]. By scanning migraine patients daily for four weeks using fMRI, functional connectivity is altered during the migraine cycle [6]. Proton magnetic resonance spectroscopy (1H-MR spectroscopy/1H-MRS) studies reported abnormal energy metabolism in migraine patients [7]. An increased concentration of glutamate + glutamine (Glx) in the occipital cortex and thalamus, increased occipital glutamate concentration [8,9], and decreased gamma-aminobutyric acid (GABA) levels in the occipital cortex [9,10] in patients with episodic migraine compared to healthy controls using 1H-MRS was demonstrated. It is assumed that an imbalance between excitatory and inhibitory neurotransmitters contributes to hyperexcitability in brain areas and thus plays a role in the development of migraine. Furthermore, 1H-MRS studies described reduced NAA levels in the occipital cortex of patients with migraine with aura, as well as decreased NAA levels in the thalamus of migraine patients without aura [7]. NAA is a marker of neuronal integrity. It is synthesized in neuronal mitochondria and reflects mitochondrial functioning [11]. It has been hypothesized that less efficient mitochondrial functioning may be predisposing to migraine [7]. Notably, the results across magnetic resonance spectroscopy (MRS) studies are not consistent, which may be because different imaging protocols were used, and that migraine is a heterogeneous disorder with individual attack frequency, absence or presence of aura, disease duration, and co-morbidities. Furthermore, MRS studies often apply different post-processing methodologies and are mainly performed in the interictal state [3]. This study aimed to characterize the course of metabolic changes and their association with functional connectivity in a patient with episodic migraine over 21 consecutive days using 1H-MRS and resting-state fMRI to obtain a better understanding of the pathophysiology of migraine and thus contribute to future improvement in diagnostics and therapy.

## 2. Materials and Methods

### 2.1. Patient

A 26-year-old male patient diagnosed with episodic migraine with aura according to the ICHD-3 participated in this study. In the three months before screening, the study participant reported 4 to 5 attacks per month, each lasting up to 24 h. The headache during a migraine attack was always localized on the left side, mainly frontal, with a pain maximum of 70–80 on a visual analogue pain scale of 0–100 (0 absence of pain, 100 most intense pain). The participant reported a visual aura, mainly flashing lights, occurring on the left side, and lasting up to 15–20 min. It first occurred shortly before diagnosis, but he did not experience an aura before every migraine attack. The participant underwent MR spectroscopy and fMRI on 21 consecutive days. He did not present any signs of neurological or other disorders and did not take regular medication of any kind. For the duration of the study, he voluntarily decided not to take any acute medication during a migraine attack. The patient was a non-smoker and did not drink any alcohol during the study period. He was asked to maintain a constant daily routine before the magnetic resonance (MR) measurements, which were always taken at the same time in the afternoon to avoid any fluctuations in the measured spectroscopic values and functional connectivity that were not migraine-related. Before each MR session, an assessment of the patient’s general condition comprising a neurologic examination, a structured headache interview, and an evaluation of premonitory signs and accompanying symptoms in detail was performed.

### 2.2. Image Acquisition

All MR measurements were performed on a Magnetom Skyra (3T, Siemens, Erlangen, Germany) using a 64-channel hydrogen head coil and standard Siemens MR sequences. A structural T1-weighted magnetization prepared rapid acquisition gradient-echo anatomical scan (MPRAGE) sequence (repetition time (TR) = 1690 ms, echo time (TE) = 2.12 ms, flip angle (FA) = 8°, field of view (FOV) = 230 × 217 mm^2^, voxel size = 0.8 isotropic) was acquired in advance for localization and for positioning of MRS grid. To clarify possible structural abnormalities, a T2-weighted Fluid-attenuated inversion recovery (FLAIR) sequence (TR = 10,000 ms, TE = 90 ms, FA = 160°, FOV = 240 × 195 mm^2^, voxel size = 0.9 × 0.9 × 3.0 mm^3^) was acquired during the initial MR measurement. Blood oxygen level-dependent (BOLD) resting-state fMRI scans were acquired using a T2-weighted gradient 2D echo-planar imaging (EPI) sequence (TR = 2400 ms, TE = 2.5 ms, FA = 80°, FOV = 240 × 233 mm, in-plane resolution = 3.5 × 3.5 mm, slices = 42 (interleaved, up-bottom), slice thickness = 3.0 mm). This was followed by two spectroscopic chemical shift imaging (CSI) sequences (TR = 1700 ms, TE = 135 ms, FOV = 120 × 120 mm^2^, FA = 90°, slice thickness = 15 mm, resolution 12 × 12, measured voxel size 10 × 10 × 15 mm^3^, interpolated resolution 16 × 16, interpolated voxel size = 7.5 × 7.5 × 15 mm^3^). For CSI standard Siemens MRI sequence “csislsr” and sequence of semi-localized by adiabatic selective refocusing (sLASER) were used. Two CSI sequences were acquired, one for the frontal brain regions, the frontal lobe and the anterior cingulate cortex (ACC); and a second for the posterior brain regions, the thalamus, the basal ganglia, and the occipital cortex (see Figure 1). The volume of interest (VOI) was planned on the isotropic 0.8 mm^3^ sagittal high-resolution multiplanar reformations (MPR) to assure the brain regions of interest are well covered for this participant with the 1H-MRS CSI grid. We obtained two CSI grids using the same TR and TE to provide optimum shimming for a good signal-to-noise ratio in these two separate brain areas. We utilized B0 and B1 shim mode “true form” (implemented in the Siemens sequence) and the water suppression method “weak water suppression” (standardized by Siemens) with a bandwidth of 50 Hz and one single measurement. The subject was instructed to keep his eyes closed for the resting-state fMRI and CSI.

The thalamus and the occipital cortex, the basal ganglia, the frontal lobe, and the ACC were selected as regions of interest for the MRS analysis; see Figure 1. These regions have been associated with changes in metabolism [8,12,13,14] and pain intensity [15] in migraine.

### 2.3. Data Processing and Analysis

#### 2.3.1. MR—Spectroscopy Analysis

The MR spectra were processed in the time domain using the advanced algorithm for accurate, robust, and efficient spectral fitting AMARES [16], which is part of the Java-based Magnetic Resonance User Interface (jMRUI) software package (v. 5.0, http://www.jmrui.eu/, accessed on 24 May 2021). [17]. The residual water line was removed by applying a Hanke/Lanczos Singular Value Decomposition (HLSVD) filter. For spectral fitting Gaussian line shapes were used. The spectrum of each voxel was visually inspected according to Kreis [18], metabolic data were evaluated for outliers, and spectra with poor quality (i.e., inadequate fitting, bad signal-to-noise ratio) were excluded completely from the analysis. Quality control was carried out visually by an experienced MR-physicist and a well-trained Radiologist and Neurologist. Prior knowledge values for estimating the metabolites were determined for the choline (Cho) peak at 3.24 parts per million (ppm), the Creatine (Cr) peak at 3.04 ppm, the Creatine 2 (Cr2) peak at 3.93 ppm, and the N-acetyl aspartate (NAA) peak at 2.02 ppm. Mean diffusivity (MD) and fractional anisotropy (FA) values were routinely calculated by the commercially available software of the MR scanner. MD and FA values were determined in the ACC, the frontal lobe, the occipital cortex, the basal ganglia, and the thalamus. In the presented study, total creatine (tCr) is defined as tCr = Cr + Cr2 [11,19]. Total creatine was used as a reference metabolite under the assumption of stable concentrations [20]. For the analysis, signal intensity ratios were generated from the spectroscopic values NAA, Cho, and tCr: NAA/tCR, NAA/Cho, and Cho/tCr. Given the exploratory nature of this study, the assessment of metabolite ratios was restricted to qualitative comparisons of relative increases or decreases in values across time. Metabolite levels were quantified separately for the left and right hemispheres. The phases of the migraine cycle were defined as follows: interictal (at least 60 h interval between the previous and the next attack), preictal (onset of headache within the next 4 h), ictal (days with a headache; postictal with headache), and postictal (24 h after the pain subsides) [2]. The preictal and postictal phases are known collectively as the peri-ictal phase. 

As we were measuring at 3 Tesla with a very high TE of 135 ms, Glx, glutamine (Gln), glutamate (Glu), taurine, and myo-inositol were not considered due to the pure signal-to-noise. For instance, Glx is regularly acquired at short TE, i.e., 30 or 40 ms, while a TE of 135 ms was used in this study. It resonates between 2.2 and 2.4 ppm chemical shifts and overlaps with the GABA peak and cannot be separated from each other [21].

#### 2.3.2. Resting-State Analysis

The resting-state data were processed using AFNI (v21.1.02, https://afni.nimh.nih.gov/, accessed on 2 Sep 2021) (afni_proc.py). Firstly, the first three volumes were discarded and despiking (3dDespike), slice timing correction (3dTshift), motion correction (3dvolreg), and alignment to a template space (align_epi_anat.py) were performed. The template image was created with ANTs’ script antsMultivariateTemplateConstruction2.sh using the 21 T1-weighted images. The template image was resampled to 2 mm isotropic resolution and a brain mask was derived (3dSkullStrp, 3dAutomask). The resting-state data were subsequently masked, scaled, and demeaned and derivatives motion estimates, as well as physiological signals captured by local white matter (ANATICOR), were regressed out. The outputs of the different steps were inspected to ensure that the data were correctly processed. Regional time courses were extracted by estimating the registration between the MNI152 space and the template image using ANTs’ (antsRegistrationSyN.sh), aligning the Harvard–Oxford subcortical brain atlas to template space, and extracting the average time course within the regions of interest (thalamus, caudate, putamen, pallidum, hippocampus, amygdala, and accumbens, both left and right). The strength of the functional connectivity between every pair of regions, for every day, was estimated by computing the Pearson’s correlation coefficients (r) between the time courses and standardizing the estimates to z-scores using the Fisher z-transformation. The pairwise connectivity strengths were compared to the concentration ratios across the 21 days using Pearson’s correlation. The results were corrected for multiple comparisons using the False Discovery Rate procedure [22] and the results were deemed significant at *p* < 0.05.

## 3. Results

Throughout the experiment, the patient experienced two migraine attacks without aura and with typical migraine accompanying symptoms such as photophobia, retreat tendency, and reinforcement of pain by physical activities. The prodromal symptoms reported by the patient were mainly fatigue and yawning. Each attack lasted for 24 h with a peak pain intensity of 50 to 85 on a visual analogue pain scale of 0 to 100. The location was frontotemporal left-sided. On four distinct days, the patient reported headaches that were not classified as migraine attacks. The localization of the latter varied between the right and left sides.

### 3.1. MR Spectroscopy Results

Presumptive migraine-associated changes were observed mainly in the concentration ratio of NAA/tCr in the left occipital lobe. Changes in NAA/tCr levels were observed before, during, and after the migraine attack. In the left-sided occipital lobe, the NAA/tCr level increased preictal and reached its peak on the first postictal day in comparison to the headache-free days (see Figure 2). The same flow was observed during the first and second attacks. Reduced Cho/tCr levels were found mainly during the first and slightly during the second migraine attack in the left basal ganglia, while peri-ictally an increase was visible.

### 3.2. Comparison of Resting-State fMRI and MR Spectroscopy

The strength of the functional connectivity between region pairs, as estimated with resting-state fMRI over 21 days, was significantly correlated with the MRS ratio NAA/Cho in the right thalamus for the right hippocampus-right pallidum (*p* = 0.019, corrected) and the right hippocampus-left pallidum (*p* = 0.006, corrected). These associations are reported in Figure 3.

## 4. Discussion

The main outcome of this study, as it is the first of its kind, is that longitudinal MR spectroscopy studies show signal changes presumably attributable to the migraine cycle. Changes in NAA levels are associated with neuronal loss and altered mitochondrial metabolism and thus abnormal energy metabolism [23]. It can be speculated that changes in NAA/tCr levels as we observed in the occipital lobe and the basal ganglia during the peri-ictal and ictal state might reflect abnormal energy metabolism and may increase the perceptivity of migraine patients to excitatory migraine triggers. These findings are consistent with the results of studies suggesting increased cerebral hyperexcitability as a trigger for a migraine attack [8,9]. Several groups could demonstrate that the basal ganglia are responsible for adaptive plasticity in the brain, affecting behavioral [24], neurological, and psychiatric conditions [25], including pain [26]. Furthermore, the basal ganglia seem to be involved in the integration of information between the cortical and thalamic regions and the three main domains of pain processing—sensory, emotional/cognitive, and endogenous/modulatory [27]. Brain imaging studies have demonstrated decreased activation in the basal ganglia of migraineurs compared to healthy controls [28] as well as increased activation in the basal ganglia during the ictal state [29]. Altered thalamic energy metabolism and functional connectivity during the peri-ictal and ictal migraine phases are in line with findings of fMRI studies. A recent fMRI study reported an increase in functional connectivity of the right thalamus with contralateral cortical areas responsible for pain modulation, e.g., the insular and orbitofrontal cortex were visible, while ipsilateral pain processing areas, e.g., the somatosensory cortex, were less connected. These findings occurred irrespective of the lateralization of the headache [5,6]. Taken together, alterations in thalamic energy metabolism and functional connectivity during a migraine attack may represent an increased modulation of nociceptive information transmitted through the thalamus. In the present study, altered levels of Cho/tCr were mainly observed in the basal ganglia. Cho is a marker of cellular membrane turnover [19]. Altered levels of Cho/tCr may reflect neuronal integrity, whereas decreased levels may suggest neuronal dysfunction [30]. It is thought that neuronal dysfunction is a predisposing factor for migraine attacks [31]. Furthermore, Cho reflects cellular proliferation as it is involved in phospholipid synthesis and degradation [19]. An increase in Cho/tCr levels is related to the abnormal proliferation of glial cells, which is a sign of neuroinflammation [32]. The inflammation might be triggered by oxidative stress or due to cortical spreading depression (CSD), both playing an important role in the pathogenesis of pain development [33,34]. In the present study, the patient was diagnosed with migraine with aura, but he did not experience any aura symptomatic during the study. It is suggested that CSD may also occur when no aura symptomatic is reported [35]. However, to discuss this topic properly further research is needed. Why different metabolic changes can be observed during the first and second migraine attacks remains elusive. It might be explained by the fact that the migraine attack itself can present differently in terms of metabolic aspects. The current study has limitations. This was a pilot study designed primarily to investigate the feasibility of using MR spectroscopy to detect metabolic changes that may be associated with the migraine cycle. One patient was included to investigate our primary endpoint. In this respect, statistical analysis is not possible. In the future, studies using longitudinal MRS with a larger cohort of patients are necessary to achieve sufficient statistical power and to obtain reproducible and statistically significant results. The severity of the disease, e.g., attack frequency, duration, as well as headache lateralization should be considered and correlated with MRS findings. Furthermore, the exact onset of the prodromal phase and the exact end of the pain cannot always be assessed with high accuracy due to clinical considerations such as sleep terminating the migraine attack.

## 5. Conclusions

In conclusion, the present pilot study indicates that in vivo longitudinal MRS has the potential to detect migraine phase-dependent changes in the brain metabolism of migraine patients. Changes in NAA/tCr and Cho/tCr were observed before, during, and after the migraine attack, mainly in the left occipital lobe and the left basal ganglia. Alterations in functional connectivity were significantly correlated with the MRS ratio NAA/Cho in the right thalamus. Furthermore, studies using MRS before and after preventive treatment are necessary to investigate whether there is a regression of the observed changes in the migraine brain after inducing a clinically relevant efficacy with preventive medication. A better understanding of changes in brain metabolism and functional connectivity of brain regions in individuals with migraine could help not only in the diagnosis, but also in therapeutic interventions.

## Figures and Tables

**Figure 1 brainsci-12-00646-f001:**
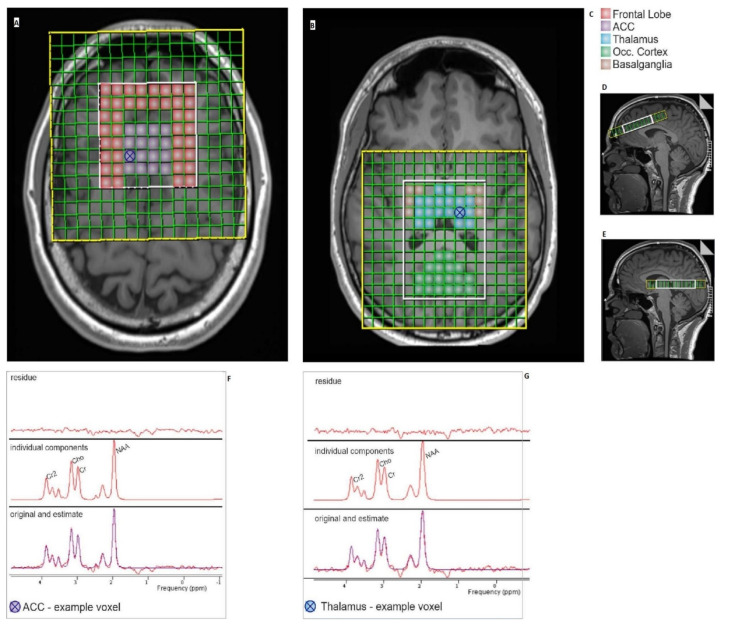
In-plane view of single layers of MRS covering: (**A**) the ACC and the frontal lobe, and (**B**) the thalamus, occipital cortex, and basal ganglia. (**C**) Colour legend of the corresponding regions. Sagittal view of the MRS layers: (**A**) top, (**B**) bottom, and (**D**,**E**) sagittal view. (⊗) A selected voxel in the (**A**) ACC and the (**B**) thalamus with corresponding MR spectra (**F**,**G**) with the peaks of N-acetyl aspartate (NAA), choline (Cho), creatine (Cr), and creatine 2 (Cr2). The height of the individual peaks of the spectra reflects their concentration; their position on the *x*-axis corresponds to the metabolite frequency in parts per million (ppm) according to the chemical shift.

**Figure 2 brainsci-12-00646-f002:**
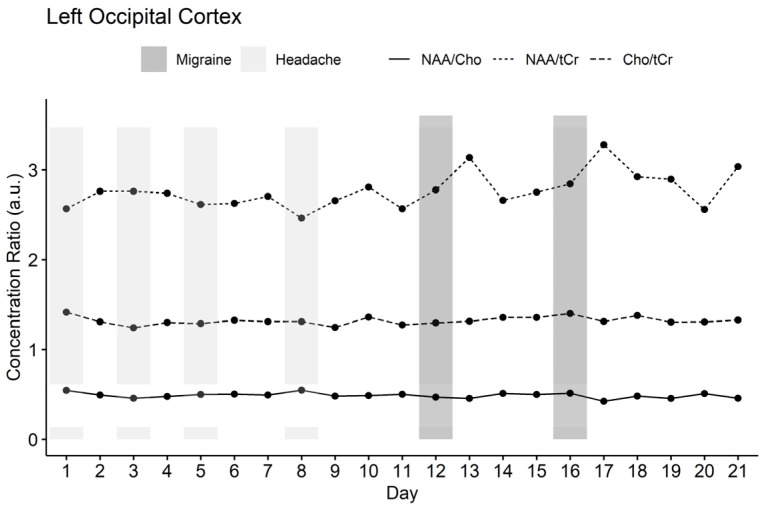
Concentration ratios of NAA/tCr, NAA/Cho, and Cho/tCr over 21 days in left-sided occipital cortex. NAA/tCr level increased preictal and reached its peak on the first postictal day in comparison to the headache-free days.

**Figure 3 brainsci-12-00646-f003:**
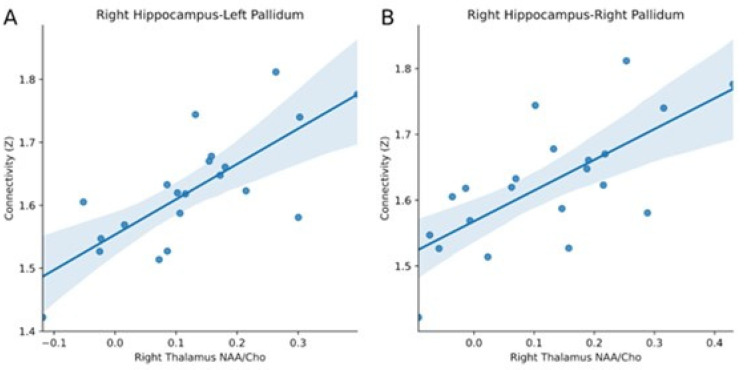
Associations between the NAA/Cho concentration in the right thalamus and the strength of the functional connectivity between (**A**) the right hippocampus-left pallidum and (**B**) the right hippocampus-right pallidum over 21 days in a migraineur.

## Data Availability

Not applicable.

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
