# Peer review of "Investigating the Migraine Cycle over 21 Consecutive Days Using Proton Magnetic Resonance Spectroscopy and Resting-State fMRI: A Pilot Study"

_brainsci, 2022, doi:10.3390/brainsci12050646_

Round 1

Reviewer 1 Report

The authors present a single case of migraine where they report 21 days of consecutive MR spectroscopy. The study is sound and the manuscript well wirtten.

I have only two minor things to address:

The present a case with MwA and also discuss about their pathophysiology. As the subject "unfortunately" only had attacks of MwoA maybe this could be skipped (it seems somewhat distracting).

I would use different "shades of grey" for figure 2, so the graph also works in bw.

Author Response

We Whave revised our manuscript to address the concerns raised by the referees. You will find a detailed point by point response to the reviewers’ comments below (Please see the attachment)

We thank the editors and reviewers for their helpful comments and hope that our revised manuscript will be considered for publication in Brain Science.

Kind regards,

Vera Filippi, Gregor Brössner

Reviewer 2 Report

The authors followed a migraineur with consecutive fMRI scans during a period of 21 days, during which two attacks of migraine without aura occurred.

NAA/tCr ratio in the left occipital lobe increased periictally compared to headache free periods during both migraine attacks whereas reduced Cho/tCr was found in the left basal ganglia.

The functional connectivity correlated with the NAA/Cho ratio in the right thalamus for the right hippocampus- right pallidum and the right hippocampus – left pallidum.

Although that there is only a single case presented, the results of this pilot study might be relevant to decipher the metabolic/functional changes in the CNS during migraine and provide a possible biomarker for the attack.

The work is interesting some minor issues should be addressed by the authors.

  1. Provide some previous contextual data about NAA, Cho in migraineurs in the introduction.
  2. In MM please describe the migraine characteristics of the patient in a more detailed way for possible further comparison, e.g. duration of the disease, average attack frequency (disease burden), the typical localization of the headache (do they appear always on the left side as during the study), the frequency and characteristics (lateralization) of the aura symptoms.
  3. Fig 2. The Cho/tCr changes, mentioned in the text, are not obvious for me looking at the figure.
  4. Fig 2 – I believe that the first sentence of the legend is incomplete (left….).
  5. Discussion: The lateralization of the MRI alterations in comparison to the headache localization should be elaborated a little bit more.

Author Response

We have revised our manuscript to address the concerns raised by the referees. You will find a detailed point by point response to the reviewers’ comments below (Please see the attachment)

We thank the editors and reviewers for their helpful comments and hope that our revised manuscript will be considered for publication in Brain Science.

Kind regards,

Vera Filippi, Gregor Brössner
